# Synergistic enhancement of AAV gene delivery in 2D cells and 3D organoids using polybrene and hydroxychloroquine

Hyeon-Jin Na[1,2☯], Yongbo Shin[2,3☯], Seung-Hyun Kim[1☯], Seung Pil Jang[1], Myung Jin Son[2,3], Yong Min Choi[1,2], Hyeon Gyeol Jeon[2,3], Ok-Seon Kwon[1,2*], Kyung-Sook Chung[1,2*]

1 Center for Gene and Cell Therapy, Korea Research Institute of Bioscience and Biotechnology, Daejeon, Republic of Korea, 2 Department of Advanced Bioconvergence, Korea University of Science and Technology, Daejeon, Republic of Korea, 3 Stem Cell Convergence Research Center, Korea Research Institute of Bioscience and Biotechnology, Daejeon, Republic of Korea

☯ These authors contributed equally to this work.
* kschung@kribb.re.kr (KSC); okskwon@kribb.re.kr (OSK)

## Abstract

Recent advances in three-dimensional (3D) culture platforms have enabled organoids to serve as physiologically relevant models for recapitulating human biology and assessing therapeutic efficacy and toxicity. Despite their promise, their complex architecture presents significant challenges for efficient gene delivery, thereby limiting their broader application in drug discovery and translational research. To overcome this challenge, we developed a sequential treatment strategy that combines polybrene (PB), which facilitates viral entry, and hydroxychloroquine (HCQ), which modulates endosomal processing. By applying PB as a pre-treatment and HCQ as a post-treatment, we achieved an approximate 1.3- to 2-fold increase in adeno-associated virus (AAV) transduction efficiency in both retinal and liver organoid models compared to single-agent treatments, and a 1.7- to 2.5-fold increases compared to treatment with virus alone. Importantly, this combinatorial treatment preserved cellular integrity, as confirmed by minimal TUNEL assay and high overall viability. Our findings demonstrate that sequential administration of PB and HCQ significantly improves AAV transduction in 3D retinal and liver organoid systems, offering a robust method to improve gene delivery. This approach not only overcomes current limitations in organoid-based research but also supports the development of more predictive platforms for evaluating AAV vectors and advancing gene therapy applications.

## Introduction

Animal models have traditionally served as the primary platform for evaluating therapeutic efficacy and safety in preclinical research. However, significant inter-species differences in immune responses and biological pathways often limit the

**Data availability statement:** All relevant data are within the manuscript and its Supporting Information files. The step-by-step experimental protocol is available at protocols.io (DOI: https://dx.doi.org/10.17504/protocols.io.4r3l29mjjv1y/v2) and also provided as Supporting Information (S1 File). The minimal dataset underlying the results of this study is provided in Supporting Information (S2 File).

**Funding:** This work was supported by the Korea Research Institute of Bioscience and Biotechnology (KRIBB) Research Initiative Program [grant number KGM5362521; Basic Science Research Program through the National Research Foundation of Korea (NRF) [grant number RS-2024-00352135]; National Research Council of Science & Technology (NST) grant by the Korea government (MSIT) (No. GTL24022-000). The funders had no role in study design, data collection and analysis, decision to publish, or preparation of the manuscript.

**Competing interests:** The authors have declared that no competing interests exist.

direct translation of these findings to human clinical applications [1]. To overcome these limitations, human organoids have emerged as a transformative platform for studying human organ development, disease pathogenesis, and pharmacological responses [2–5]. As self-organizing three-dimensional (3D) structures, organoids closely mimic the cellular heterogeneity and architecture of human tissues, thereby enabling more accurate predictions of drug efficacy, toxicity, and therapeutic outcomes [6–8]. Especially, organoid models are particularly advantageous for evaluating adeno-associated virus (AAV) tropism and transgene expression, as they provide a physiologically relevant *in vitro* platform for gene therapy research [9–13]. Despite these advantages, the multilayered structure and cellular complexity of organoids present significant barriers to the uniform delivery and expression of transgene across diverse cell populations [14]. This limitation reduces the utility of organoids in assessing cell type-specific effects of AAV-mediated gene delivery [15].

Given these challenges, enhancing AAV transduction efficiency is essential not only to achieve robust and sustained transgene expression, but also to enable accurate assessment of vector performance, dose-response relationships, and therapeutic efficacy [16]. Recent studies have investigated strategies to improve AAV-mediated gene delivery by targeting key factors in the viral life cycle, including cellular entry, intracellular trafficking, and transgene expression [17,18]. Using chemical adjuvants is a promising way to improve AAV delivery. For instance, polybrene (PB), a cationic polymer, enhances viral transduction by reducing charge repulsion between negatively charged viral particles and the cell membrane, thereby promoting more efficient viral uptake irrespective of membrane composition [19]. Similarly, hydroxychloroquine (HCQ) has been reported to increase AAV transduction by inhibiting Toll-like receptor 9 (TLR9)-mediated innate immune responses [20,21]. Although PB and HCQ individually enhance AAV transduction through distinct mechanisms, their combinatorial effect has not yet been systematically evaluated, particularly in 3D organoid systems, where structural and biological complexities present unique barriers to efficient gene delivery.

Building on these insights, this research aims to establish and optimize a combinatorial protocol for enhancing AAV delivery efficiency in human organoids through the sequential application of PB and HCQ. PB facilitates AAV cellular entry by mitigating electrostatic repulsion at the cell membrane, while HCQ improves intracellular trafficking of the viral vector by modulating endosomal processing pathways [19–21]. Given their complementary mechanisms, we demonstrated that the sequential administration of PB followed by HCQ maximizes their synergistic effects. This protocol was systematically optimized for retinal and hepatic organoid models and is expected to be broadly applicable to various human organoid systems, providing an effective strategy for improving AAV delivery in complex 3D environments. Ultimately, this combinatorial approach offers a versatile platform to improve AAV transduction efficiency across diverse organoid models, thereby advancing their utility in translational research and gene therapy development.

## Materials and methods

The protocol described in this peer-reviewed article is published on protocols.io, https://dx.doi.org/10.17504/protocols.io.4r3l29mjjv1y/v2 and is included for printing as supporting information (S1 File) with this article.

## Expected results

To validate the effects of PB and HCQ on AAV transduction, we first assessed the impact of PB in HEK293T cells. In accordance with prior studies that identified PB concentrations below 10 μg/mL as appropriate for cell-based assays, we established 10 μg/mL for 4 hours as the working condition [22,23]. Validation of this dosage was achieved through time-dependent CCK-8 assays, which consistently yielded ≥80% cell viability up to 48 hours (S1A Table), supporting minimal cytotoxicity. Under these conditions, cells were treated with PB (10 μg/mL) for 4 hours prior to infection with an mCherry-expressing AAV at a multiplicity of infection (MOI) of $2 \times 10^4$. PB treatment led to a marked increase in mCherry fluorescence intensity compared to untreated controls (Fig 1A). Bright-field imaging confirmed that PB did not adversely affect cell morphology or density, indicating minimal cytotoxicity. Flow cytometry performed 2 days post-transduction revealed that both the proportion of mCherry-positive cells and the level of transgene expression were significantly higher in the PB-treated group than in the virus-only group ($^{****}p < 0.0001$ and $^{**}p < 0.01$, respectively; Fig 1B and 1C). Overall, PB treatment enhanced AAV transduction efficiency by approximately 1.53-fold (from $15.7 \pm 1.7\%$ to $24.0 \pm 1.6\%$) (Fig 1B). These results demonstrate that PB serves as a simple effective enhancer of AAV-mediated gene delivery in HEK293T cells.

We next investigated the impact of HCQ on AAV transduction. Based on previous reports indicating that effective HCQ concentrations in cell-based assays are within the range of 3–18.75 μM, we selected 15 μM for 1 hour as the working condition [20]. Validation of this dosage was achieved through time-dependent CCK-8 assays, which consistently yielded ≥80% cell viability up to 48 hours (S1B Table). HEK293T cells were pretreated with HCQ (15 μM) for 1 hour prior to infection with an mCherry-expressing AAV at an MOI of $2 \times 10^4$. HCQ pretreatment significantly increased mCherry fluorescence intensity compared to controls (Fig 2A). Consistent with PB treatment, bright-field imaging showed no adverse changes in cell morphology or density, indicating minimal cytotoxicity. Flow cytometry analysis conducted 2 days post-transduction revealed that both the percentage of mCherry-positive cells and the intensity of transgene expression were significantly elevated in the HCQ-treated group relative to the virus-only group ($^{****}p < 0.0001$ and $^{**}p < 0.01$, respectively; Fig 2B and 2C). Overall, HCQ pretreatment resulted in more than a 2.16-fold increase (from $21.1 \pm 3.2\%$ to $45.6 \pm 0.3\%$) in transduction efficiency while maintaining cell viability above 90% (Fig 2B). These results are consistent with previous reports suggesting that HCQ enhances AAV intracellular trafficking, likely by modulating endosomal processing.

Given that PB enhances viral entry and HCQ improves intracellular processing of viral particles, we hypothesized that their sequential administration—PB as a pre-treatment followed by HCQ as a post-treatment—would synergistically enhance AAV-mediated gene delivery by targeting distinct stages of the transduction pathway. To evaluate this, liver and retinal organoids were generated following established protocols and validated at defined maturation stages prior to application of the sequential treatment protocol. [9,24–26]

First, liver organoids were pretreated with PB (8 μg/mL) or HCQ (20 μM) for 1 hour prior to AAV transduction, with additional HCQ post-treatment (20 μM for 36 hours) applied to selected groups. Confocal imaging on day 10 post-transduction showed increased mCherry fluorescence in both PB- and HCQ-treated groups compared to untreated controls (Fig 3A). Quantitative image analysis showed a 1.29- to 1.65-fold increase in transgene expression with single-agent treatment, relative to the virus-only group, and a 1.97- to 1.27-fold increase with sequential treatment (PB + HCQ or HCQ + HCQ), relative to the respective single-agent groups (Fig 3B). Compared to the virus-only group, sequentially treated organoids exhibited a 2.54- (from $12.3 \pm 2.8\%$ to $31.4 \pm 2.7\%$) to 2.11-fold increase (from $12.3 \pm 2.8\%$ to $26.0 \pm 2.3\%$) in the

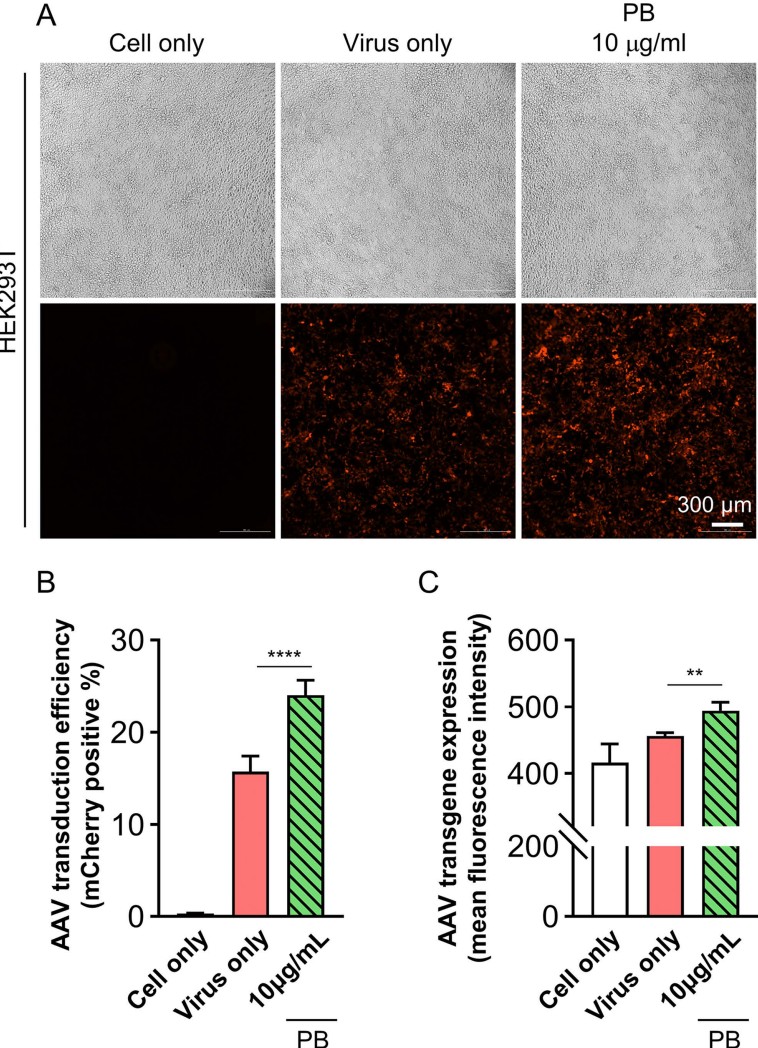

**Fig 1. Enhanced AAV transduction efficiency by polybrene (PB) in HEK293T cells.** (A) Representative microscopic images of HEK293T cells on day 2 post-transduction. PB was treated at a concentration of 10 µg/ml for 4 hours before AAV transduction. The virus was applied at a multiplicity of infection MOI of $2 \times 10^4$. Scale bar: 300 µm. (B), (C) Quantitative analysis of HEK293T on day 2 post-transduction. Statistical significance was determined using one-way ANOVA followed by Tukey's multiple comparisons test. **$p < 0.01$, ****$p < 0.0001$ indicate significant differences compared to the Virus-only group.

mCherry-positive area. Notably, no significant changes in organoid morphology or compactness were observed under any treatment condition. TUNEL assay performed on day 10 revealed a low proportion of TUNEL-positive cells (1.1–3.9%), with no statistically significant differences compared to controls (Fig 3C and 3D; S1 Fig). Validation of the TUNEL staining was achieved by strong nuclear signals in DNase I-treated positive controls.

Next, retinal organoids were pretreated with PB (8 µg/mL for 4 hours) or HCQ (15 µM for 1 hour) prior to AAV transduction, with additional HCQ post-treatment (15 µM for 48 hours) applied to selected groups. Confocal imaging on day 15 post-transduction demonstrated a significant increase in mCherry fluorescence specifically in the sequential PB + HCQ group, whereas PB or HCQ alone, as well as the HCQ + HCQ sequential treatment, did not exhibit notable enhancement compared to the virus-only control (Fig 4A). Quantitative image analysis further confirmed a 1.68-fold increase

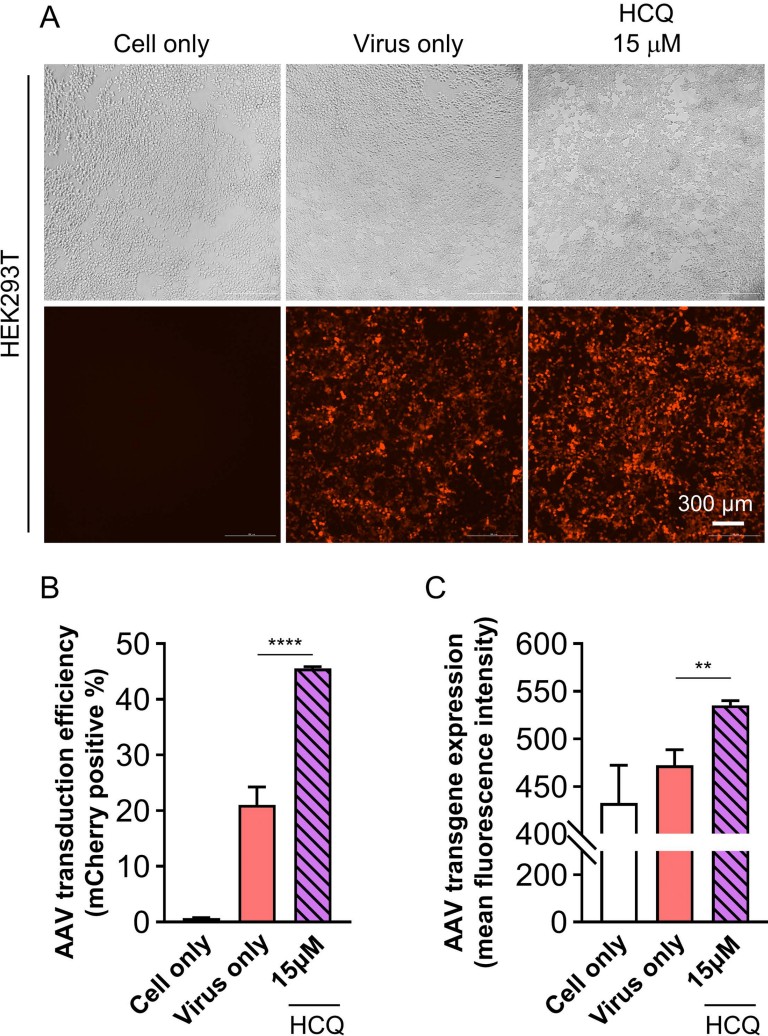

**Fig 2. Hydroxychloroquine (HCQ) enhances AAV-mediated gene delivery in HEK293T cells.** (A) Representative microscopic images of HEK293T cells on day 2 post-transduction. HCQ was treated at a concentration of 15 µM for 1 hour before AAV transduction. The virus was applied at an MOI of $2 \times 10^4$. Scale bar: 300 µm. (B), (C) Quantitative analysis of HEK293T cells on day 2 post-transduction. Statistical significance was determined using one-way ANOVA followed by Tukey's multiple comparisons test. **$p < 0.01$, ****$p < 0.0001$ indicate significant differences compared to the Virus-only group.

(from 17.9 ± 3.5% to 30.0 ± 8.4%) in mean fluorescence intensity in the PB + HCQ group relative to the virus-only group (**$p < 0.01$), while no statistically significant improvement was observed in the other treatment groups (Fig 4B). Morphological assessment at day 15 revealed well-preserved structural integrity across all treatment conditions, with no detectable abnormalities. TUNEL staining similarly indicated a low frequency of TUNEL-positive nuclei (1.2–2.2%) across all conditions, with no significant differences observed (ns, $p \geq 0.05$; Fig 4C and 4D; S2 Fig).

Collectively, these results indicate that sequential PB and HCQ treatment exerts a synergistic effect, offering a reliable and well-tolerated strategy for enhancing AAV-mediated gene delivery in both liver and retinal organoids. This protocol, which leverages mechanistically complementary actions in a sequential manner, is broadly applicable across diverse organoid systems and highlights its potential as an efficient delivery strategy for organoid-based gene therapy research.

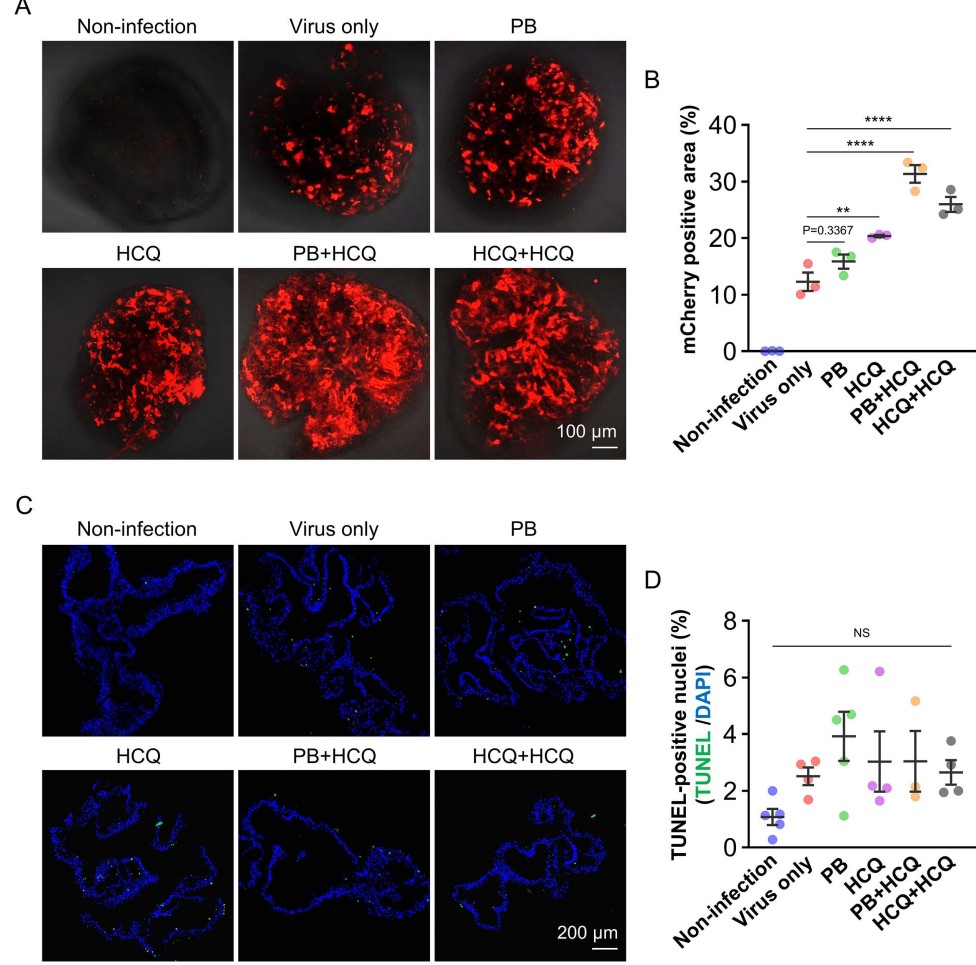

**Fig 3. AAV transduction efficiency in liver organoids with PB and HCQ treatment.** (A) Representative fluorescence images of liver organoids on day 10 post-transduction. For single treatments (PB or HCQ), organoids were pretreated with 8 µg/mL PB or 20 µM HCQ for 1 hour prior to AAV transduction. For sequential treatments (PB + HCQ and HCQ + HCQ), organoids were pretreated with 8 µg/mL PB or 20 µM HCQ for 1 hour, followed by post-treatment with 20 µM HCQ for 36 hours. Scale bar: 100 µm. (B) Quantification of mCherry-positive area in AAV-transduced liver organoids using ImageJ. Data are presented as mean ± SEM (n = 3). (C) Representative fluorescence images of TUNEL staining (green) in liver organoids on day 10 post-transduction. Scale bar: 200 µm. (D) Quantification of TUNEL-positive nuclei as a percentage of total DAPI-stained nuclei using ImageJ. Data are presented as mean ± SEM (n = 3–5). Statistical significance was determined using one-way ANOVA followed by Tukey's multiple comparisons test. **p < 0.01, ****p < 0.0001 indicate significant differences compared to the Virus-only group.

## Discussion

Chemical enhancement of AAV transduction efficiency using PB and HCQ is achieved by following the protocol established in this study, supported by our results. In 3D organoid models, the synergistic effects of PB and HCQ can provide an effective platform for evaluating AAV transduction. In this section, we provide troubleshooting guidance aimed at ensuring the consistency and reproducibility of the protocol. Several technical considerations should be taken into account to ensure reproducibility of this protocol.

First, optimization of AAV MOI is critical for the assay system, as excessively high titers of AAV can induce cytotoxicity. In our study, liver organoids were transduced with AAV8 across a range of $10^4$–$10^6$ vg/cell and retinal organoids with $1 \times 10^{10}$ vg/organoid to determine the optimal concentration, although further adjustment may be required depending on the AAV serotype or organoid type.

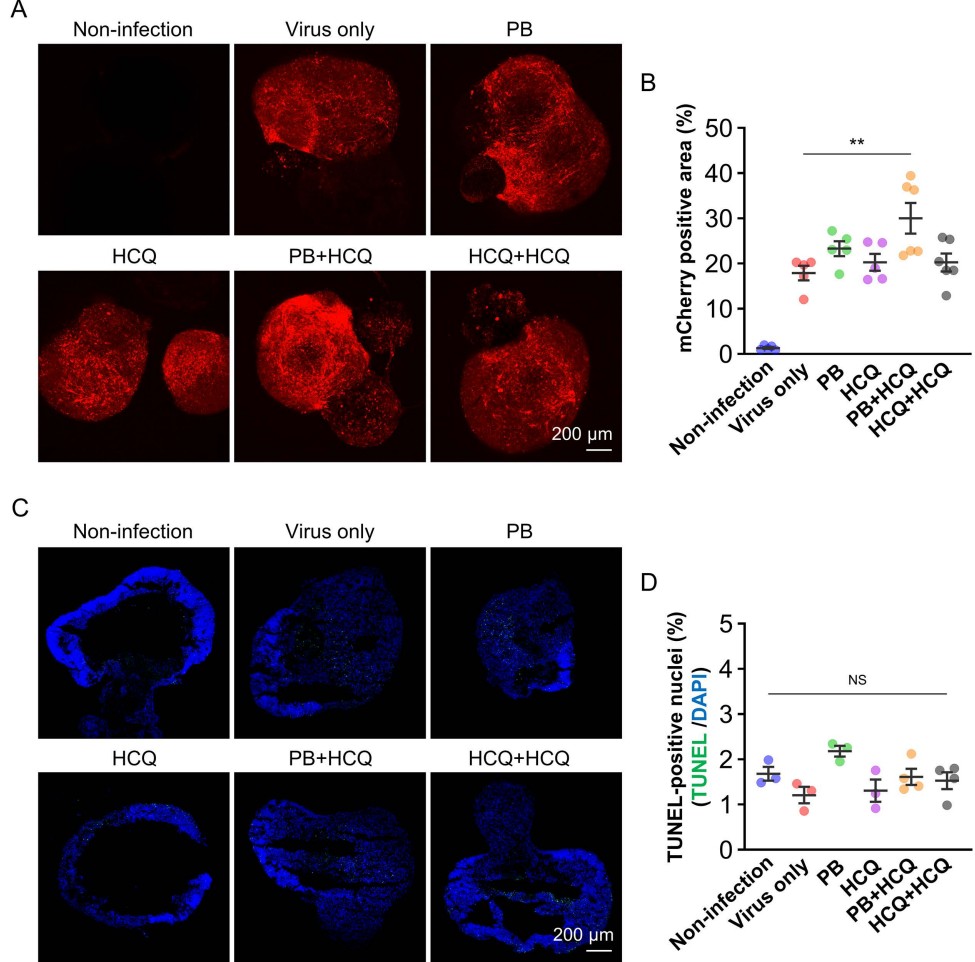

**Fig 4. Enhanced AAV transduction with preserved viability in retinal organoids treated by PB and HCQ. (A)** Representative fluorescence images of retinal organoids on day 15 post-transduction. PB was applied at 10 µg/mL for 4 hours and HCQ at 15 µM for 1 hour prior to AAV transduction. For the PB + HCQ group, organoids were pretreated with PB for 4 hours, followed by post-treatment with HCQ for 48 hours. For the HCQ + HCQ group, organoids were pretreated with HCQ for 1 hour, followed by a second HCQ treatment for 48 hours post-transduction. The virus was applied at an MOI of $1 \times 10^{10}$. Scale bar: 200 µm. **(B)** Quantification of mCherry-positive area in AAV-transduced retinal organoids using ImageJ. Data are presented as mean ± SEM (n = 5–6). **(C)** Representative fluorescence images of TUNEL staining in retinal organoids on day 15 post-transduction. Scale bar: 200 µm. **(D)** Quantification of the TUNEL-positive area was performed using ImageJ. Data are presented as mean ± SEM (n = 3–4). Statistical significance was determined using one-way ANOVA followed by Tukey's multiple comparisons test. **$p < 0.01$ indicates a significant difference compared to the Virus-only group; ns ($p \geq 0.05$) indicates no significant difference.

Next, determining the appropriate concentrations of PB and HCQ is important to balance efficacy and cytotoxicity. For PB, previous studies employing AAV as well as retroviral and lentiviral vectors have shown that concentrations below 10 µg/mL are commonly used and are associated with improved transduction efficiency [22,23]. Similarly, previous studies have shown that HCQ promotes AAV transduction in a dose-dependent manner up to 18.75 µM, at which point the effect reaches a threshold [20]. Cell viability assays demonstrated that the PB and HCQ concentrations selected for our AAV transduction experiments maintained acceptable levels of cell viability for up to 48 hours, supporting the suitability of these conditions throughout the experimental time course.

Finally, variability in organoid quality, including differences in size, density, and maturation stage, represents a critical determinant of transduction outcomes. To minimize this variability, organoids should be validated prior to viral

transduction. For example, liver organoids can be confirmed by expression of hepatic markers such as HNF4A, BSEP, and ALB at both RNA and protein levels, along with functional secretion of ALB and AAT in culture supernatants [26]. Similarly, retinal organoids can be validated by the expression of canonical retinal markers such as PAX6, CRX, and RCVRN, as well as by structural features including distinct neuroepithelial layering [9]. Defining acceptable size ranges (e.g., 400–600 μm in diameter for liver organoids; 800–1000 μm for mature retinal organoids) provides an additional quality control measure to ensure consistent transduction outcomes across experiments. Together, these troubleshooting considerations provide practical guidance for adapting the protocol across laboratories and organoid systems.

## Supporting information

**S1 File. Step-by-step protocol.** Step-by-step protocol, also available on protocols.io
(PDF)

**S2 File. Minimal dataset underlying the results of this study.**
(XLSX)

**S1 Table. Minimal dataset of PB and HCQ dose- and time-response viability measurements.** Cell viability of HEK293T cells treated with PB (10 μg/mL, A) or HCQ (15 μM, B) was measured at the indicated time points using the CCK-8 assay. Data are presented as mean ± SD relative to untreated controls (0 hr, set as 100%; n = 4). Statistical significance was determined using one-way ANOVA with Tukey's multiple comparisons test; all treatment groups showed $p < 0.0001$ compared to control, as denoted by the "&" symbol in the table.
(DOCX)

**S1 Fig. High-resolution TUNEL assay images of liver organoids.** TUNEL staining was performed on liver organoids at day 10 post-transduction. As a positive control, organoids were treated with DNase I prior to staining. TUNEL-positive cells are shown in green, and nuclei are counterstained with DAPI (blue). Magnified views of the areas outlined by white dashed boxes are shown at the bottom of each panel. White arrowheads indicate TUNEL-positive nuclei. Scale bars: 200 μm (top), 50 μm (middle), 10 μm (bottom).
(TIF)

**S2 Fig. High-resolution TUNEL assay images of retinal organoids with DNase I treated positive control.** Positive control images of DNase I-treated retinal organoids confirm the validity of the TUNEL assay. Enlarged (40×) images highlight the co-localization of DAPI-stained cell nuclei with TUNEL-positive signals. Magnified views of the regions indicated by white dashed boxes are displayed at the bottom of each panel. Scale bars: 200 μm (top), 100 μm (middle), 5 μm (bottom).
(TIF)

## Acknowledgments

We appreciate the assistance of the KOBIC Research Support Program.

## Author contributions

**Conceptualization:** Kyung-Sook Chung.

**Data curation:** Hyeon-Jin Na, Yongbo Shin, Seung Pil Jang, Kyung-Sook Chung.

**Formal analysis:** Hyeon-Jin Na, Yongbo Shin.

**Funding acquisition:** Ok-Seon Kwon, Kyung-Sook Chung.

**Investigation:** Hyeon-Jin Na, Yongbo Shin, Hyeon Gyeol Jeon.

**Methodology:** Hyeon-Jin Na, Yongbo Shin, Seung-Hyun Kim.

**Project administration:** Ok-Seon Kwon, Kyung-Sook Chung.

**Resources:** Yong Min Choi, Hyeon Gyeol Jeon.

**Software:** Seung-Hyun Kim.

**Supervision:** Seung Pil Jang, Kyung-Sook Chung.

**Validation:** Hyeon-Jin Na, Yongbo Shin.

**Visualization:** Seung-Hyun Kim, Hyeon Gyeol Jeon.

**Writing – original draft:** Hyeon-Jin Na, Yongbo Shin.

**Writing – review & editing:** Seung Pil Jang, Myung Jin Son, Ok-Seon Kwon, Kyung-Sook Chung.

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
