## [Decision Letter · Decision Letter 0]

18 Aug 2025

Dear Dr. Chung,

We look forward to receiving your revised manuscript.

Kind regards,

Xiaoping Bao, Ph.D.

Academic Editor

PLOS ONE

Additional Editor Comments (if provided):

Reviewers' comments:

Reviewer's Responses to Questions

**Comments to the Author**



Reviewer #1: Yes

Reviewer #2: Yes

2. Has the protocol been described in sufficient detail?

To answer this question, please click the link to protocols.io in the Materials and Methods section of the manuscript (if a link has been provided) or consult the step-by-step protocol in the Supporting Information files.

Reviewer #1: Partly

Reviewer #2: Partly

3. Does the protocol describe a validated method?

Reviewer #1: Yes

Reviewer #2: Yes

4. If the manuscript contains new data, have the authors made this data fully available?

Reviewer #1: Yes

Reviewer #2: Yes

**5. Is the article presented in an intelligible fashion and written in standard English?**

Reviewer #1: Yes

Reviewer #2: Yes

Reviewer #1: This laboratory protocol presents a straightforward strategy for boosting adeno associated virus (AAV) transduction in both 2D adherent cultures and 3D organoids through sequential treatment with polybrene (PB) and hydroxychloroquine (HCQ). The rationale is clear, the data are easy to follow, and the approach should be of practical value to investigators performing routine viral transduction. To maximize the impact and reproducibility of the protocol, however, several issues need to be addressed:

- The manuscript claims enhanced transduction in retinal and liver organoids, yet provides no molecular or functional validation of these organoids. Please include standard characterization data and relevant functional to confirm that the structures used truly represent the target tissues.

- Please expand the protocol to include complete routine culture and passaging instructions for each cell type tested, especially H9 hESCs.

- Please provide a step‑by‑step description (media compositions, growth factors, timing, seeding densities, and critical quality‑control checkpoints) for both retinal and liver organoid generation. As written, the procedure is too abbreviated for replication. If the methods are based on previously published approaches, please cite the corresponding references in the manuscript while still including sufficient details within the protocol to ensure reproducibility.

- The materials table is comprehensive; however, many reagents listed are not referenced in the protocol due to insufficient procedural details. Please incorporate all relevant information into the protocol text to ensure the method can be easily and accurately reproduced.

- If applicable, please add a concise troubleshooting section that flags common pitfalls (e.g., PB‑induced cytotoxicity, HCQ precipitation, organoid size variability) and offers solutions or alternative conditions.

Reviewer #2: This is an interesting protocol addressing an important challenge in AAV transduction, particularly in 3D culture systems. The approach of testing multiple chemical additives and their combination has the potential to be highly useful to the field. However, several aspects of the methodology would benefit from additional clarification and data to strengthen reproducibility and user guidance. My main comments are as follows.

1. In the combination experiments, the chemical concentration differs from the single compound tests. Please explain the rationale such as synergy and toxicity mitigation, and provide viability/toxicity data for each condition to guide in reproducing the protocol effectively.

2. Imaging timepoints differ (48 h VS 72 h) across conditions. Please specify whether these were optimized individually and how users should select the appropriate timepoints when applying the protocol.

3. Because some compounds may alter proliferation or viability, include viability at each timepoints.

4. Using different vg/cell for entry assays and for expression analysis may mask improvements in transduction efficiency, as this MOI approaches saturation in HEK cells. Including an intermediate MOI (10^3~10^5 vg/cell) in the expression assay would allow assessment of combined entry and expression effect under non-saturating conditions.

5. The protocol specifies a range of concentrations for the chemical treatments, but does not provide sufficient explanation of guidance on optimizing them for different experimental contexts. Including the rationale such as dose-response data or toxicity considerations and recommendations for optimization would greatly enhance reproducibility.

**Do you want your identity to be public for this peer review?** For information about this choice, including consent withdrawal, please see our Privacy Policy

Reviewer #1: No

Reviewer #2: No

---

## [Author Response · Author response to Decision Letter 1]

8 Oct 2025

Title: Synergistic enhancement of AAV gene delivery in 2D cells and 3D organoids using polybrene and hydroxychloroquine

Dear Xiaoping Bao,

We sincerely thank you and the reviewers for the time and effort invested in evaluating our manuscript entitled “Synergistic enhancement of AAV gene delivery in 2D cells and 3D organoids using polybrene and hydroxychloroquine.” We highly appreciate the constructive comments and insightful suggestions provided, which have been invaluable in improving the quality and clarity of our work.

In this revised version, we have carefully addressed all the reviewers’ comments point by point. We performed additional experimental analyses and provided clarifications where requested, and we believe these revisions have significantly strengthened the manuscript. Detailed responses to each comment are included in below.

Sincerely,

Kyung-Sook Chung, Ph.D.

Director, Principal Investigator

Center for Gene and Cell Therapy, KRIBB, Daejeon 34141, Republic of Korea

Tel: (82) 42-879-8200

Fax: (82) 42-860-4597

E-mail: kschung@kribb.re.kr

The following are our detailed responses to the reviewers’ comments:

Reviewer #1, Comment 1

1. The manuscript claims enhanced transduction in retinal and liver organoids, yet provides no molecular or functional validation of these organoids. Please include standard characterization data and relevant functional to confirm that the structures used truly represent the target tissues.

Response: We appreciate the reviewer for emphasizing the importance of validation. To address this concern, we have added references to previously validated protocols for retinal and liver organoid generation (Ref 9, 23-25). These well-established protocols are considered to support the consistency of organoid formation. By citing these established works, we provide validation of the organoids used in our study and ensure that their biological relevance is supported by published data. This point has now been clarified in the revised manuscript(Lines 125-127)

Reviewer #1, Comment 2

2. Please expand the protocol to include complete routine culture and passaging instructions for each cell type tested, especially H9 hESCs.

Response: We appreciate the reviewer for these helpful suggestions and agree that adding culture details and linking the listed reagents to the corresponding steps will improve reproducibility. Accordingly, we have updated the linked protocol on protocols.io to include routine culture and passaging instructions for H9 hESCs, and we have ensured that all reagents listed in the materials table are explicitly referenced in the step-by-step procedures. The revised protocols.io entry and corresponding updated PDF have been uploaded with this submission, and the manuscript now references this updated version.

Reviewer #1, Comment 3

3. Please provide a step by step description (media compositions, growth factors, timing, seeding densities, and critical quality control checkpoints) for both retinal and liver organoid generation. As written, the procedure is too abbreviated for replication. If the methods are based on previously published approaches, please cite the corresponding references in the manuscript while still including sufficient details within the protocol to ensure reproducibility.

Response: We appreciate the reviewer’s suggestion for highlighting the importance of methodological validation. As mentioned in response to comment 1 above, the retinal and liver organoids used in this study were generated following our previously established protocols (Ref 9, 23-25). These protocols have been characterized in terms of molecular and functional markers. We have now cited these references in the revised manuscript to emphasize that the organoids employed in our study are based on well-validated and reproducible methodologies.

Reviewer #1, Comment 4

4. The materials table is comprehensive; however, many reagents listed are not referenced in the protocol due to insufficient procedural details. Please incorporate all relevant information into the protocol text to ensure the method can be easily and accurately reproduced.

Response: We thank the reviewer for this helpful comment and for highlighting the importance of clear linkage between the materials list and the experimental procedures. We fully agree that accurate referencing of all reagents is essential for reproducibility. In our protocols.io entry, all reagents listed in the Materials table are incorporated into the relevant procedures. Specifically, most of the reagents are used in the preparation of organoid media, and their detailed formulations are provided in the Preparation of Medium part within the Materials. We believe this structure ensures that each reagent can be clearly traced to its experimental use, thereby facilitating accurate reproduction of the method.

Reviewer #1, Comment 5

5. If applicable, please add a concise troubleshooting section that flags common pitfalls (e.g., PB induced cytotoxicity, HCQ precipitation, organoid size variability) and offers solutions or alternative conditions.

Response: We appreciate the reviewer for this constructive suggestion and agree that including troubleshooting guidance enhances the practical utility of the protocol. Accordingly, we have incorporated a troubleshooting note into the Discussion of the manuscript (Lines 181-205). In this section, we outline common challenges, such as optimization of viral input, PB- and HCQ-induced cytotoxicity, and variability in organoid size. For example, we note that excessively high titers of AAV can induce cytotoxicity, PB toxicity can be mitigated by adjusting concentration or exposure time, HCQ efficacy and toxicity must be balanced through careful dose and duration selection, and organoid size variability can be reduced by standardizing seeding density and monitoring maturation checkpoints. We believe these additions will provide practical guidance for researchers applying this protocol.

Reviewer #2, Comment 1

1. In the combination experiments, the chemical concentration differs from the single compound tests. Please explain the rationale such as synergy and toxicity mitigation, and provide viability/toxicity data for each condition to guide in reproducing the protocol effectively.

Response: We appreciate the reviewer for this thoughtful comment and agree that clear justification of concentration choices is important for reproducibility. We would like to clarify that the combination experiments were conducted only in organoids, and within each organoid system the concentrations for single and combined treatments were identical.

For retinal organoids, PB was used at 10 μg/mL and HCQ at 15 μM, while for liver organoids PB was used at 8 μg/mL and HCQ at 20 μM. These concentrations are within the ranges reported in previously published protocols (Ref 20, 26) and were applied consistently across the respective experiments. We hope that this explanation addresses the reviewer’s concern and provides reassurance regarding the consistency of our approach.

Reviewer #2, Comment 2

2. Imaging timepoints differ (48 h VS 72 h) across conditions. Please specify whether these were optimized individually and how users should select the appropriate timepoints when applying the protocol.

Response: We appreciate the reviewer for this important comment and agree that consistency in imaging timepoints is essential for reproducibility. To ensure clarity and uniformity across the protocol, we have standardized all analyses to the 48 h timepoint, which provides a reliable window for detecting AAV transduction. Accordingly, the main figures in the revised manuscript have been updated to reflect this consistent timepoint. (Lines 87-101, Fig 1)

Reviewer #2, Comment 3

3. Because some compounds may alter proliferation or viability, include viability at each timepoints.

Response: We appreciate the reviewer for this constructive comment and for emphasizing the importance of reproducibility and user guidance. We fully agree that including viability data is essential to help users interpret treatment effects and apply the protocol with confidence. To address this, we performed additional viability assays and included the results as a new Supporting Information (S4 File). These results indicate that the concentrations selected for our AAV transduction experiments maintain acceptable levels of cell viability throughout the experimental time course. We have also incorporated this information into the linked protocol on protocols.io so that future users can readily apply the protocol under comparable conditions.

Reviewer #2, Comment 4

4. Using different vg/cell for entry assays and for expression analysis may mask improvements in transduction efficiency, as this MOI approaches saturation in HEK cells. Including an intermediate MOI (10^3~10^5 vg/cell) in the expression assay would allow assessment of combined entry and expression effect under non-saturating conditions.

Response: We appreciate the reviewer for this insightful comment and agree that avoiding saturating conditions is important when assessing transduction efficiency. We would like to clarify that the MOIs used in HEK cells, we used intermediate MOI (2×104 vg/cell), which already falls within the range suggested by the reviewer and does not represent a saturating condition. We hope this clarification resolves any potential confusion regarding the MOIs applied in our assays.

Reviewer #2, Comment 5

5. The protocol specifies a range of concentrations for the chemical treatments, but does not provide sufficient explanation of guidance on optimizing them for different experimental contexts. Including the rationale such as dose-response data or toxicity considerations and recommendations for optimization would greatly enhance reproducibility.

Response: We appreciate the reviewer for this important comment. The concentrations of PB and HCQ used in our assays were selected based on previously published studies, and we have cited these references in the revised manuscript to clarify this rationale. Furthermore, we have incorporated a troubleshooting note into the Discussion section (Lines 181–208) that provides additional context on dose considerations, including guidance for potential optimization. We believe this guidance will enhance reproducibility and provide clearer instructions for adapting the protocol to different experimental settings.

---

## [Decision Letter · Decision Letter 1]

21 Oct 2025

Synergistic enhancement of AAV gene delivery in 2D cells and 3D organoids using polybrene and hydroxychloroquine

PONE-D-25-35275R1

Dear Dr. Chung,

We’re pleased to inform you that your manuscript has been judged scientifically suitable for publication and will be formally accepted for publication once it meets all outstanding technical requirements.

Kind regards,

Xiaoping Bao, Ph.D.

Academic Editor

PLOS ONE

Additional Editor Comments (optional):

Reviewers' comments:

Reviewer's Responses to Questions

**Comments to the Author**



Reviewer #1: Yes

2. Has the protocol been described in sufficient detail?

To answer this question, please click the link to protocols.io in the Materials and Methods section of the manuscript (if a link has been provided) or consult the step-by-step protocol in the Supporting Information files.

Reviewer #1: Yes

3. Does the protocol describe a validated method?

Reviewer #1: Yes

4. If the manuscript contains new data, have the authors made this data fully available?

Reviewer #1: Yes

**5. Is the article presented in an intelligible fashion and written in standard English?**

Reviewer #1: Yes

Reviewer #1: The revisions and point-by-point responses adequately address all reviewer comments and improve the clarity and completeness of the manuscript.

**Do you want your identity to be public for this peer review?** For information about this choice, including consent withdrawal, please see our Privacy Policy

Reviewer #1: No

---

## [Editor Report · Acceptance letter]

PONE-D-25-35275R1

PLOS ONE

Dear Dr. Chung,

I'm pleased to inform you that your manuscript has been deemed suitable for publication in PLOS ONE. Congratulations! Your manuscript is now being handed over to our production team.

Kind regards,

on behalf of

Dr. Xiaoping Bao

Academic Editor

PLOS ONE